# Assessing Healthcare Workers’ Knowledge and Their Confidence in the Diagnosis and Management of Human Monkeypox: A Cross-Sectional Study in a Middle Eastern Country

**DOI:** 10.3390/healthcare10091722

**Published:** 2022-09-08

**Authors:** Malik Sallam, Kholoud Al-Mahzoum, Ala’a B. Al-Tammemi, Mohammed Alkurtas, Fatemeh Mirzaei, Nariman Kareem, Hala Al-Naimat, Laila Jardaneh, Laith Al-Majali, Akram AlHadidi, Khaled Al-Salahat, Eyad Al-Ajlouni, Nadin Mohammad AlHadidi, Faris G. Bakri, Harapan Harapan, Azmi Mahafzah

**Affiliations:** 1Department of Pathology, Microbiology and Forensic Medicine, School of Medicine, The University of Jordan, Amman 11942, Jordan; 2Department of Clinical Laboratories and Forensic Medicine, Jordan University Hospital, Amman 11942, Jordan; 3Department of Translational Medicine, Faculty of Medicine, Lund University, 22184 Malmö, Sweden; 4School of Medicine, The University of Jordan, Amman 11942, Jordan; 5Migration Health Division, International Organization for Migration (IOM), The UN Migration Agency, Amman 11953, Jordan; 6Department of Pathology, Al-Kindy College of Medicine, University of Baghdad, Baghdad 00964, Iraq; 7Student Research Committee, Golestan University of Medical Sciences, Gorgan 49138-15739, Iran; 8School of Dentistry, The University of Jordan, Amman 11942, Jordan; 9Department of Internal Medicine, School of Medicine, The University of Jordan, Amman 11942, Jordan; 10Infectious Diseases and Vaccine Center, The University of Jordan, Amman 11942, Jordan; 11Medical Research Unit, School of Medicine, Universitas Syiah Kuala, Banda Aceh 23111, Indonesia; 12Tropical Disease Centre, School of Medicine, Universitas Syiah Kuala, Banda Aceh 23111, Indonesia; 13Department of Microbiology, School of Medicine, Universitas Syiah Kuala, Banda Aceh 23111, Indonesia

**Keywords:** public health emergency of international concern, health professional, global health emergency, attitude, vaccination, MPXV, epidemic

## Abstract

The ongoing multi-country human monkeypox (HMPX) outbreak was declared as a public health emergency of international concern. Considering the key role of healthcare workers (HCWs) in mitigating the HMPX outbreak, we aimed to assess their level of knowledge and their confidence in diagnosis and management of the disease, besides the assessment of their attitude towards emerging virus infections from a conspiracy point of view. An online survey was distributed among HCWs in Jordan, a Middle Eastern country, during May–July 2022 using a questionnaire published in a previous study among university students in health schools in Jordan. The study sample comprised 606 HCWs, with about two-thirds being either physicians (*n* = 204, 33.7%) or nurses (*n* = 190, 31.4%). Four out of the 11 HMPX knowledge items had <50% correct responses with only 33.3% of the study respondents having previous knowledge that vaccination is available to prevent HMPX. A majority of study respondents (*n* = 356, 58.7%) strongly agreed, agreed or somewhat agreed that the spread of HMPX is related to a role of male homosexuals. Confidence in the ability of diagnosis based on the available monkeypox virus diagnostic tests was reported by 50.2% of the respondents, while the confidence levels were lower for the ability to manage (38.9%) and to diagnose (38.0%) HMPX cases based on their current level of knowledge and skills. Higher confidence levels for HMPX diagnosis and management were found among physicians compared to nurses. The endorsement of conspiracy beliefs about virus emergence was associated with lower HMPX knowledge, the belief in the role of male homosexuals in HMPX spread, and with lower diagnosis and management confidence levels. The current study highlighted the gaps in knowledge regarding HMPX among HCWs in Jordan as well as the lack of confidence to diagnose and manage cases among physicians and nurses. Raising the awareness about the disease is needed urgently considering the rapid escalation in the number of cases worldwide with reported cases in the Middle East. The attitude towards male homosexuals’ role in HMPX spread necessitates proper intervention measures to prevent stigma and discrimination among this risk group. The adoption of conspiratorial beliefs regarding virus emergence was widely prevalent and this issue needs to be addressed with proper and accurate knowledge considering its potential harmful impact.

## 1. Introduction

On 23 July 2022, the World Health Organization (WHO) Director-General declared the ongoing human monkeypox (HMPX) multi-country outbreak as a public health emergency of international concern (PHEIC) [1,2]. This statement came in light of monkeypox virus (MPXV) rapid spread to many non-endemic countries for the first time, with an evident risk of further expansion internationally [3].

Between May and July 2022, more than 18,000 HMPX cases were identified in over 75 countries spanning the six WHO world regions [4]. Nevertheless, unequal distribution of disease toll was observed with 70% of the cases being confirmed in the European region and 25% in the region of the Americas [5,6]. As of 9 August 2022, the total number of reported cases reached 31,425 in 82 countries that has not historically reported HMPX, with 375 cases in seven endemic countries [7]. By 31 August 2022, the total number of confirmed HMPX cases approached 50,000 in 99 countries/territories worldwide [8]. Thirty-eight HMPX cases have been confirmed in six Arab countries of the Middle East and North Africa (United Arab Emirates (UAE): 16, Saudi Arabia: 8, Lebanon: 6, Morocco: 3, Qatar: 3 and Sudan: 2 cases), with no recorded cases in Jordan by the end of August 2022 [8].

Prior to the ongoing HMPX outbreak, this emerging virus infection was limited to the endemic regions in West and Central Africa, with rare MPXV spillover events linked to travel to endemic areas or animal imports from these regions [9,10,11,12,13].

The causative agent of HMPX is the MPXV classified within the genus *Orthopoxvirus* of the *Poxviridae* family [14]. Variola virus (VARV) is the other well-known lethal poxvirus that is classified with MPXV in the *Orthopoxvirus* genus [14]. Shared characteristics between the two viruses confer similarities in clinical features and cross-protection following recovery or vaccination [15,16].

The disease can be transmitted either by direct contact (with skin or mucous membrane lesions or via the respiratory tract/saliva), or indirectly through contaminated fomites [17,18]. Following an incubation period ranging between 5 and 21 days, the onset of HMPX involves a prodrome of flu-like symptoms followed by the development of cutaneous manifestations evolving from macules to papules, vesicles, pustules and scabs [6,17,19]. However, a number of unusual presentations were reported in the course of the current 2022 outbreak including penile swelling and rectal pain requiring hospital admission [20]. Although HMPX can be considered a self-limited disease, complications could occur involving bacterial superinfection, respiratory distress, encephalitis and dehydration [6,21]. The case-fatality ratio has been previously reported at 0–11%; however, the HMPX-related mortalities were extremely low during the ongoing outbreak [5,22].

Although anyone in close contact with active cases can be at risk of virus acquisition, the current cases appear to be concentrated among men who have sex with men (MSM) [18,23,24,25,26]. This striking pattern was evident in a recent study on HMPX in London which showed that 196 out of 197 (99.5%) reported cases were identified among MSM (gays and bisexual men) [20]. A recent study from Spain added further evidence supporting the hypothesis of skin-to-skin transmission of MPXV during sex as the dominant mode of transmission among MSM in the course of the ongoing outbreak [18]. This transmission clustering pattern among MSM might be attributed to highly interconnected social and sexual networks; however, this explanation is pending further studies similar to the recent study by Akira Endo et al. [27]. Due to the predominant reporting of HMPX among MSM, a special attention should be paid to tackle issues of stigmatization and discrimination which can be as dangerous as the virus itself [1,28].

Public health measures to prevent forward transmission of MPXV focus on spreading knowledge and awareness especially among most-at-risk groups; however, the central preventive measure relies on vaccination [17,29,30]. It has been previously shown that smallpox vaccination provides at least 85% effectiveness to prevent HMPX; therefore, vaccination of most-at-risk groups has already been considered [16].

An important aspect that appears to accompany the emergence of virus infections with subsequent control measures is the circulation of rumors and misinformation. This was evident during the ongoing coronavirus disease 2019 (COVID-19) pandemic and its negative impact included the association with higher anxiety levels besides the higher rates of vaccination hesitancy [31,32]. Conspiracy ideas emerged early on during the current HMPX outbreak including the belief that the virus was bioengineered for a political cause and the belief in 5G network role in virus spread [33]. Thus, the impact of such conspiratorial ideas should be evaluated to assess its impact particularly on health seeking behavior [34,35].

Healthcare workers (HCWs) are a key group to be considered for focused knowledge and awareness to be prepared to provide proper responses especially during outbreaks and emergence of novel infectious diseases [36]. The role of HCWs is crucial to identify the cases for early isolation, and for vaccinating close contacts for containment and mitigation. Thus, prompt assessment of their preparedness can be highly valuable as an initial step similar to the study conducted by Matteo Riccò et al. involving Italian physicians [29].

In our previous study that investigated HMPX knowledge among university students in Jordanian health schools, unsatisfactory levels of knowledge were found, besides its association with embrace of conspiracy beliefs towards emerging viral infections [37]. In this study, we aimed to evaluate the level of knowledge regarding HMPX among HCWs in Jordan, as well as to evaluate their confidence levels in the diagnosis and management of the disease with a special focus on physicians and nurses. Additionally, the objectives of this study involved the investigation of the correlation between HMPX knowledge and the endorsement of conspiracy beliefs about emerging viruses.

## 2. Materials and Methods

### 2.1. Study Design

The current cross-sectional study was based on the distribution of an online self-administered questionnaire to assess HMPX knowledge and confidence in diagnosis and management of the disease among HCWs in Jordan. The occupational categories that fit our definition of HCWs included: physicians, nurses, dentists, pharmacists and medical technicians (laboratory, radiology, rehabilitation, and anesthesia technicians).

Accordingly, the questionnaire was distributed online using the Google Forms tool during 25 May 2022–8 July 2022 (Appendix A). The questionnaire was prepared and distributed in the Arabic language without incentives for participation. Response to all items was mandatory to reduce the item non-response bias.

Sampling was based on chain-referral starting with the contacts of the authors in Jordan with announcements of the survey on the following social media platforms: Facebook, Twitter and Instagram together with the following instant messaging applications: WhatsApp and Messenger.

Calculation of the minimum sample size was based on the latest estimates regarding the number of HCWs in Jordan (~100,000 HCWs) [38]. The minimum required sample size was 598 based on an estimated proportion of 0.5 with 4% as the desired precision of estimate and 95% confidence level [39].

The study was approved by the Scientific Research Committee at the School of Medicine—University of Jordan (Reference No. 2544/2022/67). An electronic informed consent was obtained by the presence of a mandatory item at the introductory section of the questionnaire “Do you agree to participate in this study?”. If the respondent answered “No” then the survey form was closed immediately.

### 2.2. Survey Instrument

The questionnaire used in this study was developed based on the previously published studies by Harapan et al. for HMPX knowledge and confidence items and by Freeman et al. for emerging virus infections conspiracy scale (EVICS) that was adopted in our previous study addressing HMPX knowledge and attitude among university students in health schools in Jordan [37,40,41,42]. The complete survey items were presented comprehensively in our previous study among students in health schools/faculties in Jordan [37].

Briefly, following the introductory and informed consent section, sociodemographics were assessed involving age, sex, place of residence, educational level and occupational category. This was followed by HMPX knowledge section using eleven items with yes/no/I do not know as the possible responses. These eleven items were used to calculate the HMPX Knowledge Score (HMPX K-Score) as a sum of score for each item in each respondent, with correct response scored as one, incorrect response as minus one, while I do not know was scored as zero.

Confidence in diagnosis and management of HMPX was assessed using three items with yes/no as the possible responses [41]. Confidence score was calculated as the sum of the score of the three items for each respondent with yes scored as one and no scored as zero. The confidence score showed a Cronbach’s alpha value of 0.759.

Attitude towards the male homosexuals’ role in the spread of HMPX was assessed by a single item using a seven-point Likert scale (strongly agree to strongly disagree through neutral/no opinion). The use of “male homosexuals” term instead of MSM was based on the more common usage of the former term in Jordan.

Finally, a 12-item section was included to assess conspiracy beliefs regarding emerging virus infections. Each item was rated using a seven-point Likert scale (strongly agree scored as 7, agree scored as 6, somewhat agree scored as 5, neutral/no opinion scored as 4, somewhat disagree scored as 3, disagree scored as 2 and strongly disagree scored as 1). Thus, a high EVICS score value implied the endorsement of conspiracy beliefs regarding emerging virus infections [37]. The complete questionnaire in English is provided in (Appendix A).

### 2.3. Study Measures

The primary outcome variables were: (1) HMPX knowledge based on HMPX K-Score (dichotomized based on the mean value into HMPX K-Score ≤ 4 indicating inferior knowledge vs. HMPX K-Score > 4 suggestive of better knowledge); (2) Confidence in HMPX diagnosis and management based on the confidence score dichotomized into two categories: lower confidence with a score of zero or one vs. higher confidence scored as two or three; (3) Attitude towards the role of male homosexuals in the spread of HMPX divided into two categories (agreement/neutral attitude involving those who strongly agreed, agreed or somewhat agreed in male homosexuals’ role besides those with neutral or had no opinion vs. disagreement attitude involving those who strongly disagreed, disagreed or somewhat disagreed); and (4) Conspiratorial attitude towards emerging virus infections was dichotomized as EVICS ≥ 48 indicating higher belief in conspiracy vs. EVICS < 48 indicating less endorsement of such beliefs.

### 2.4. Statistical Analysis

The analysis was conducted in IBM SPSS Statistics for Windows, Version 22.0. Armonk, NY, USA: IBM Corp. Chi-squared test (χ^2^), Mann–Whitney *U* test (M-W), Kruskal Wallis test (K-W) and multinomial regression analysis were used as appropriate. The statistical significance level was determined at *p* < 0.050.

## 3. Results

### 3.1. Characteristics of the Study Participants

The number of Jordanian HCWs that comprised the final study sample was 606, divided as follows: physicians (*n* = 204, 33.7%), nurses (*n* = 190, 31.4%), pharmacists (*n* = 74, 12.2%), medical technicians (*n* = 70, 11.6%) and dentists (*n* = 68, 11.2%). The general characteristics of the study respondents stratified per occupational category are shown in (Table 1). The following variables predominated in the study sample: females (*n* = 368, 60.7%), respondents with an undergraduate degree (*n* = 450, 74.3%) and respondents residing in the Capital, Amman (*n* = 368, 60.7%). The mean age for the whole study sample was 35 years (median = 34, standard deviation = 9).

### 3.2. Human Monkeypox Knowledge among Jordanian HCWs

The overall level of HMPX knowledge per item among the study respondents is illustrated in (Figure 1). Only four out of the 11 HMPX knowledge items had >70% correct responses, with four items having <50% correct responses. Only 33.3% of the study respondents knew that vaccination is available to prevent HMPX.

For the HMPX K-Score, the mean was 4.2 (range: −4 to 11). A significantly higher HMPX K-Score was found among males (mean = 4.4 vs. 4.0 in females, *p* = 0.041, M-W), and respondents with a postgraduate degree (mean = 4.7 vs. 4.0 among those with undergraduate degrees, *p* = 0.007, M-W), while age and place of residence did not show statistically significant differences based on HMPX K-Scores. For occupational categories, medical technicians and physicians had higher HMPX K-Score (mean = 4.7 and 4.6, respectively) compared to pharmacists, nurses and dentists (mean = 4.2, 3.7 and 3.4, respectively, *p* = 0.002, K-W).

### 3.3. The Attitude of the Study Respondents towards the Role of Male Homosexuals in HMPX Spread

More than half of the study respondents strongly agreed, agreed or somewhat agreed that the spread of HMPX is related to the role of male homosexuals (*n* = 356, 58.7%). Upon comparing all variable categories to attitude towards the role of male homosexuals in HMPX spread item divided into three categories (agree vs. neutral vs. disagree) no statistically significant differences were found (Figure 2).

### 3.4. Confidence among Jordanian HCWs to Diagnose and Manage HMPX Cases

About a half of the study respondents were confident in their ability of HMPX diagnosis based on the available MPXV diagnostic tests in their facilities (*n* = 304, 50.2%). The confidence levels were lower for the ability to manage and diagnose HMPX cases based on their current level of knowledge and skills (*n* = 236, 38.9% and *n* = 230, 38.0%, respectively).

Subsequent confidence analysis involved physicians and nurses only considering their role in diagnosis and management of HMPX cases. A higher confidence score was found among physicians compared to nurses (mean = 1.56 vs. 1.22, *p* = 0.005, M-W) and among respondents with HMPX K-Score >4 (mean = 1.60 vs. 1.23 among those with HMPX K-Score ≤ 4, *p* = 0.002, M-W), while age (*p* = 0.408), sex (*p* = 0.143), residence (*p* = 0.442) and educational level (*p* = 0.798) did not show statistically significant differences. Multinomial regression analysis with confidence score as the dependent variable, HMPX K-Score and occupational category (physicians vs. nurses) as the factors with age, sex, educational level and place of residence as the covariates showed that higher knowledge was significantly correlated with higher confidence (Table 2).

### 3.5. The Determinants of Conspiracy Beliefs Regarding Emerging Virus Infections

The overall attitude of the study respondents towards emerging virus infections conspiracies was neutral (mean = 47.4, median = 48.0, interquartile range = 32.0–61.0). Per item, the highest mean value was found for the items “I am skeptical about the official explanation regarding the cause of virus emergence”, with a mean value of 4.5 out of seven and the item “Viruses are biological weapons manufactured by the superpowers to take global control”, with a mean value of 4.3 out of seven, while the lowest mean was found for the item “I do not trust the information about the viruses from scientific experts” with a mean value of 3.5 out of seven (Figure 3).

Higher EVICS scores indicative of higher embrace of conspiracies regarding emerging virus infections were found among females (mean = 50.5 vs. 44.1 among males, *p* < 0.001, M-W), respondents aged ≥ 34 years (mean = 49.6 vs. 45.1, *p* = 0.002, M-W), respondents with inferior HMPX knowledge (mean = 49.3 vs. 45.0, *p* = 0.003, M-W), nurses and medical technicians compared to physicians, dentists and pharmacists (mean = 53.5 vs. 50.6 vs. 41.4 vs. 44.3 vs. 47.8, *p* < 0.001, K-W), and respondents who believed in the role of male homosexuals in HMPX spread (mean = 51.1 vs. 45.6 in those with neutral/no opinion response vs. 38.0 in those who disagreed with this opinion, *p* < 0.001, K-W). The differences were not statistically significant for the educational level (*p* = 0.101, M-W) and residence (*p* = 0.116, M-W).

Multinomial logistic regression analysis showed that EVICS ≥ 48 was independently correlated with inferior HMPX knowledge and with the agreement or neutral/no opinion belief in the role of male homosexuals in HMPX spread, while physicians showed the opposite pattern (with medical technicians as the reference group, Table 3).

## 4. Discussion

The timely and proper response of HCWs is an important pre-requisite to challenge the ongoing HMPX outbreak. To achieve this goal, it is necessary to assess the baseline level of HMPX knowledge especially among physicians and nurses considering their central role in patient care. This approach can help to raise awareness of HMPX and to conduct well-informed training among HCWs. Consequently, this can strengthen HCWs’ preparedness for mitigation and response to face the emerging threat that HMPX poses as advocated by the WHO [43]. Amid the ongoing HMPX outbreak, HCWs represent a key group which can be at risk of disease acquisition and forward transmission of the virus. Therefore, the evaluation of their knowledge about the disease and their confidence levels to diagnose and manage the possible cases is of utmost value. Subsequently, the gaps in knowledge can be addressed with proper education and training. Additionally, providing proper knowledge and training can improve the quality of care among HCWs which is an essential target in safe and high quality patient service [44]. A few past and recent studies had similar objectives and we aimed to follow their path in assessing HMPX knowledge among HCWs in Jordan [29,40,41,45,46].

Four major findings were the highlights of this study which can be listed as follows: (1) Low level of HMPX knowledge was found among HCWs in Jordan, and it was associated with lower confidence in management and diagnosis among physicians and nurses, besides its association with a higher embrace of conspiracy beliefs about emerging virus infections; (2) More than a half of the study respondents across all tested variables agreed either strongly, or at least to some extent that male homosexuals had a role in the spread of HMPX and this view was independently associated with a higher embrace of conspiracy beliefs regarding emerging virus infections; (3) The occupational category was an evident determinant of HMPX knowledge besides the endorsement of conspiracies regarding emerging virus infections; and (4) Although the overall attitude towards conspiratorial ideas regarding emerging virus infections was neutral, a considerable proportion of the study participants endorsed conspiracy beliefs with skepticism towards the official explanation of virus emergence and the belief that virus emergence could be a part of a biological warfare.

In this study, the overall level of knowledge regarding HMPX among HCWs in Jordan was unsatisfactory for a majority of the tested items. Specifically, only four out of the 11 knowledge items were correctly identified by more than 70% of the respondents. This pointed to satisfactory knowledge regarding awareness that HMPX is caused by a virus, cutaneous manifestations of the disease and the shared features with smallpox. Similar levels of knowledge about these aspects was previously identified in a study conducted among general practitioners in Indonesia prior to the current HMPX outbreak [40]. On the other hand, discernable defects in knowledge appeared in relation to different aspects of the disease including clinical presentation, epidemiology, transmission and prevention. For example, only a third of the study respondents correctly knew that vaccination is available to prevent HMPX.

The issue of potential need for MPXV vaccination requires a special attention following the COVID-19 pandemic, during which vaccination hesitancy emerged as a major hindrance to the efforts needed to control the pandemic, especially in the Middle East region [47]. Currently, vaccination to prevent HMPX is not available in Jordan [48]. However, vaccination might be needed soon at least for most-at-risk groups considering the spread of the virus to neighboring countries and at the global level [7]. Thus, the assessment of HCWs’ attitude to HMPX vaccination is recommended soon. A recent study among Italian physicians showed a slightly favorable attitude towards MPXV vaccination [29]. The result was consistent with recent reports that displayed generally positive attitude of HCWs towards vaccination in Italy [49,50,51]. However, more studies with similar aims are encouraged to inform vaccination policies among HCWs worldwide, since timely results are needed to protect HCWs considering their frontline position facing the HMPX outbreak and to take into account the regional differences in vaccine acceptance that were observed during COVID-19 pandemic [32,52].

Additional results of the study in relation to HMPX knowledge included the differences in the level of knowledge based on occupational category. Specifically, higher HMPX knowledge was observed among physicians compared to nurses. Considering the association of HMPX knowledge to confidence in management and diagnosis, the minor yet significant differences should be considered through adjusting the education and training to meet the differences observed for each profession [53].

Compared to our previous study among university students in health schools in Jordan, the overall knowledge score was slightly higher in this study. This result can be fathomable for several reasons. First, the possible interest of HCWs in HMPX based on their direct role in patient care. Second, the rapid escalation of cases of this reemerging disease with its potential threats. Third, the extended period of the survey distribution in this study. Finally, the increased media coverage and rapid availability of literature tackling different aspects of HMPX recently [6,17,20,23,25,54]. Despite the widespread media coverage of the evolving status of HMPX, less than a half of the study respondents knew about the multi-country outbreak. This can be attributed to the lack of cases in Jordan with scarcity of the reported cases in the Middle East region and also to the late declaration of HMPX as a PHEIC.

Possible explanations for the relatively low level of HMPX knowledge reported in this study can be the young mean age of the study sample (35 years), translating into a majority of respondents living post smallpox eradication era with declining focus on poxviruses in education and training [6,29,40,41]. This involves reduced coverage in medical and other healthcare-related curricula [40]. Moreover, prior to the current outbreak, monkeypox cases were rare with limited human-to-human transmission with subsequent lack of attention regarding the disease outside the endemic countries [13].

Despite the limited literature that studied knowledge, attitude and practice (KAP) of HCWs on HMPX, our findings were in line with the studies from Indonesia and Italy [29,40]. In the study by Harapan et al., that was conducted in 2019, less than 20% of the general practitioners had heard of HMPX during their medical education which highlights the lack of HMPX education in academic curriculum [40]. A recent HMPX KAP study by Matteo Riccò et al. reported similar low levels of knowledge among Italian physicians [29]. Low levels of HMPX knowledge were also reported among the general public in Saudi Arabia, Lebanon and Kurdistan region of Iraq [55,56,57].

Improving knowledge and increasing the awareness about the disease can be valuable especially in relation to adherence to infection control measures [58]. Gaps in knowledge regarding MPXV transmission were found in this study as well as in the recent study among Italian physicians, as well as among health professionals in Kuwait [29,46].

In our study, the HMPX diagnosis and management confidence levels were generally low among physicians and nurses. This can be attributed to the relatively young age of the study sample with subsequent lower level of confidence in medical practice as reported previously by Harapan et al. in the context of confidence to manage HMPX among general practitioners in Indonesia [41]. However, in contrast to the aforementioned study, our findings showed that higher HMPX knowledge was significantly associated with higher self-reported confidence levels in HMPX diagnosis and management. Therefore, improving knowledge can be reflected in better boosting confidence which in turn can be helpful in better response to the current outbreak. Nevertheless, generally low levels of confidence were found in this study particularly for clinical diagnosis and management of HMPX. The study by Harapan et al. suggested that, attending national conferences (at least one) and receiving data during medical education helps with better confidence acquisition [41].

In this study, a majority of participants agreed that MSM had a role in the spread of the ongoing HMPX outbreak. This result was uniform across all sociodemographic and occupational categories. Although this finding does not necessarily imply a negative attitude towards MSM stricto sensu, it should be evaluated further especially in the Middle East region, where a few studies pointed to the generally negative attitude towards MSM in the context of HIV infection, besides the reporting of social stigma and bullying towards COVID-19 patients in the early phases of the pandemic [59,60,61,62]. This issue needs further evaluation in future studies, since in the words of the WHO Director General “stigma and discrimination can be as dangerous as any virus” [1].

In this study, the embrace of conspiracy beliefs regarding emerging virus infections was independently associated with agreement or lack of opinion towards the role of male homosexuals in HMPX spread. To the best of our knowledge, no previous studies assessed the correlation between conspiratorial ideas (particularly those related to emerging viral infection) and the stigmatizing attitude towards most-at-risk group. Therefore, research is needed in this area especially in the Middle East where both conspiracy theories and stigma/discrimination towards most-at-risk group for HMPX (namely MSM) is highly prevalent [59,60].

The issue of stigma and discrimination following an epidemic and directed towards the risk groups is reminiscent of the early days of AIDS description in the early 1980s. Specific conspiracy beliefs were previously shown to be negatively correlated with health behavior and with stigma and discrimination. For example, conspiracy beliefs regarding HIV/AIDS was found to be a barrier to preventive efforts of the disease in relation to medical mistrust [63]. Recently, a study that was conducted in Belgium and France showed a harmful effect of conspiracy mentality reflected in less trust in medical and scientific institutions [64]. Furthermore, genocidal HIV beliefs was linked to less public support and participation in HIV-related programs as shown by Laura Bogart et al. in South Africa [65].

Finally, this study evaluated the embrace of conspiratorial ideas regarding emerging virus infections including the beliefs in official explanation of virus emergence, its possible link to biological warfare, and the attitude towards subsequent intervention measures including lockdowns implemented during the COVID-19 pandemic. Consistent with our previous study among university students in health schools in Jordan, a substantial proportion of HCWs in this study endorsed conspiracy beliefs regarding emerging virus infections [37]. The roots of conspiratorial ideas regarding virus origin appear old but gained momentum during COVID-19 pandemic [66]. The findings in this study revealed the high prevalence of adoption of these beliefs. Although such ideas might appear harmless, several studies have shown its negative impact including the association with high anxiety levels, negative impact on public health intervention measures and the high rates of vaccination hesitancy [32,67,68]. This area of research is important considering the immediate circulation of misinformation during COVID-19 pandemic and the current HMPX outbreak. Furthermore, the relevance of this aim is related to the almost immediate circulation of misinformation and conspiracy theories about the origin of HMPX [33,69,70,71].

The study limitations included selection bias based on the sampling approach, with a limited sample size besides the inherent social desirability bias in relation to EVICS items. In addition, the future studies should assess the attitude towards vaccination and the possible stigmatizing attitude towards MSM considering their high risk of virus acquisition [72]. Moreover, the current study was limited by the lack of items assessing seniority, specialty, career grading, and other relevant qualifications of HCWs’ categories similar to the approach taken by Matteo Riccò et al. [29]. Such items are needed in the future work aiming to discern differences within each occupational category in terms of HMPX knowledge and confidence levels in diagnosis and management of the disease. Furthermore, the evaluation of HCWs’ attitude towards the role of MSM in HMPX spread was based on a single item, with subsequent risk of measurement bias. Therefore, we encourage future studies focusing on the prevalence of HCWs’ stigmatizing attitude towards HMPX patients, and particularly towards most-at-risk groups using a more rigorous approach. Finally, addressing the potential hesitancy towards HMPX vaccination and its associated determinants should be considered in future studies [29].

## 5. Conclusions

Unsatisfactory levels of HMPX knowledge were found among HCWs in Jordan. This can hamper the efforts needed to identify and manage HMPX cases and should be tackled by proper and timely awareness and educational courses, workshops and alerts.

The results of the current study can help to catalyze well-informed and coordinated efforts to help in HCWs’ preparedness to face the potential threats of HMPX outbreak. The differences in knowledge per occupational category hint to the need of awareness programs tailored to each health profession. In addition, improving HMPX knowledge can have a beneficial impact on improving the HCWs’ confidence to diagnose and manage the disease considering the low levels of confidence observed among physicians and nurses in this study. The potential negative impact of endorsing conspiracy beliefs in relation to virus emergence requires proper action, which can be done through providing adequate knowledge considering its vital role in changing attitude and behavior. This is particularly important for social media platforms which are frequently used as the source of information about emerging virus infections [55].

## Figures and Tables

**Figure 1 healthcare-10-01722-f001:**
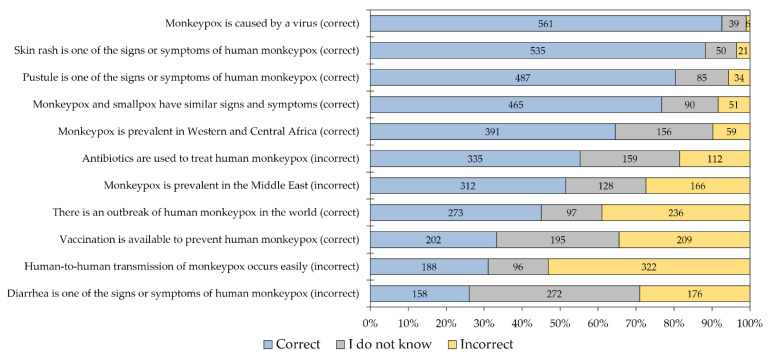
The overall level of human monkeypox knowledge per item among the study respondents.

**Figure 2 healthcare-10-01722-f002:**
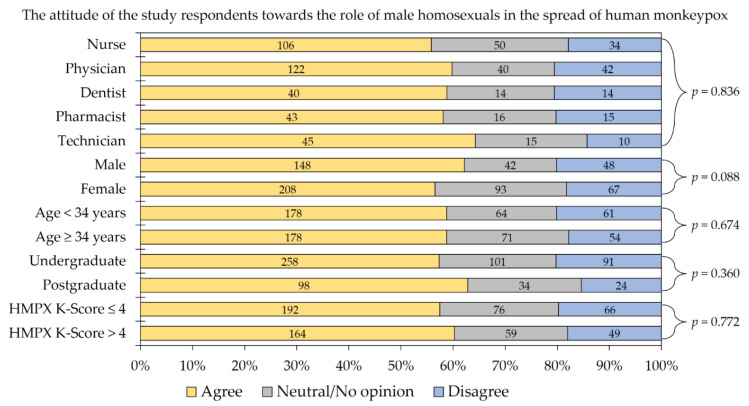
The attitude of the study respondents towards the role of male homosexuals in the spread of human monkeypox (HMPX). The “Agree” category involved the following responses combined: strongly agree, agree and somewhat agree. The “Disagree” category involved the following responses combined: strongly disagree, disagree and somewhat disagree. HMPX K-Score: human monkeypox knowledge score. *p* values were calculated using the chi-squared test.

**Figure 3 healthcare-10-01722-f003:**
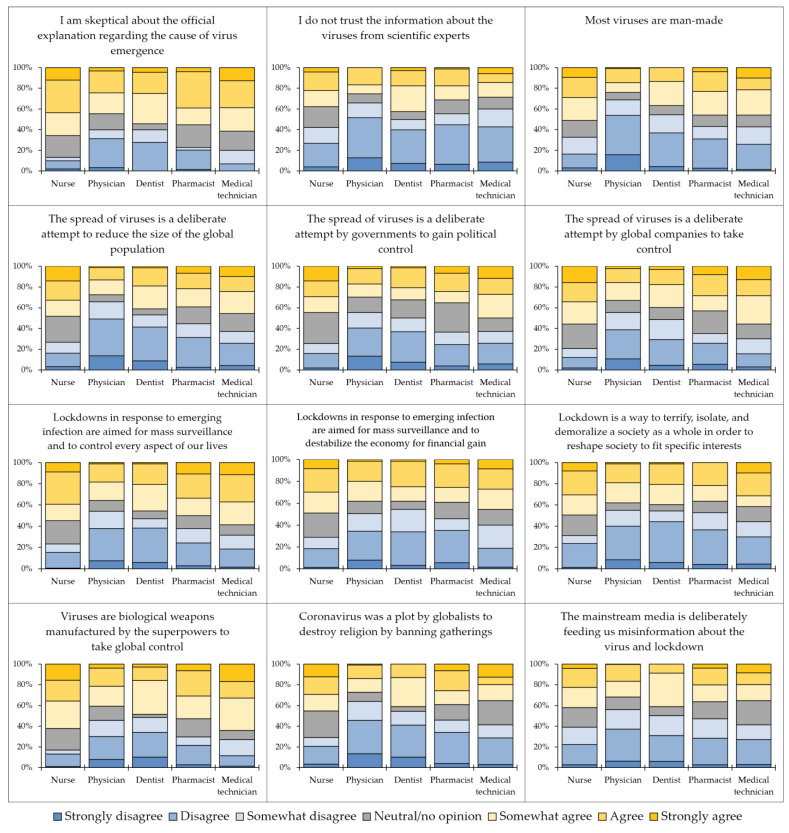
The attitude towards emerging virus infections conspiracy beliefs items in among healthcare workers who participated in this study divided by occupational category.

**Table 1 healthcare-10-01722-t001:** The overall features of the study respondents stratified per occupational category.

Variable	Group	Occupational Category *n* (%)				
		Nurse (*n* = 190)	Physician (*n* = 204)	Dentist (*n* = 68)	Pharmacist (*n* = 74)	Medical Technician ^2^ (*n* = 70)
Age in years (mean ± SD ^1^, median)		39 ± 9, 39	33 ± 10, 29	32 ± 8, 30	35 ± 9, 33	35 ± 9, 34
Age	<34 years	49 (25.79)	140 (68.63)	43 (63.24)	39 (52.70)	32 (45.71)
	≥34 years	141 (74.21)	64 (31.37)	25 (36.76)	35 (47.30)	38 (54.29)
Sex	Male	44 (23.16)	129 (63.24)	28 (41.18)	25 (33.78)	12 (17.14)
	Female	146 (76.84)	75 (36.76)	40 (58.82)	49 (66.22)	58 (82.86)
Educational level	Undergraduate	158 (83.16)	139 (68.14)	51 (75.00)	58 (78.38)	44 (62.86)
	Postgraduate	32 (16.84)	65 (31.86)	17 (25.00)	16 (21.62)	26 (37.14)
Residence	The Capital (Amman)	85 (44.74)	137 (67.16)	49 (72.06)	49 (66.22)	48 (68.57)
	Outside the Capital	105 (55.26)	67 (32.84)	19 (27.94)	25 (33.78)	22 (31.43)

^1^ SD: Standard deviation; ^2^ Medical Technician: Medical technicians including professions in laboratory, radiology, rehabilitation, and anesthesia.

**Table 2 healthcare-10-01722-t002:** Factors correlated with higher confidence in diagnosis and management of human monkeypox (HMPX) among physicians and nurses who participated in the study (*n* = 394).

Factors Associated with Higher Confidence in Diagnosis and Management of HMPX ^1^	Odds Ratio (95% Confidence Interval)	*p* Value
HMPX K-Score ^2^ > 4 vs. HMPX K-Score ≤ 4	1.575 (1.044–2.378)	0.030
Nurses vs. physicians	0.724 (0.441–1.188)	0.201
Covariates		
Age < 34 years vs. ≥34 years	1.029 (0.654–1.619)	0.902
Males vs. females	0.927 (0.595–1.446)	0.739
Undergraduates vs. postgraduates	0.851 (0.526–1.379)	0.513
Residence in Amman vs. outside Amman	0.776 (0.509–1.182)	0.238

^1^ The confidence score dichotomized as those with a score of zero or 1 (lower confidence) vs. those with a score of 2 or 3 (higher confidence) with the higher confidence as the reference category. ^2^ HMPX K-Score: Human monkeypox knowledge score.

**Table 3 healthcare-10-01722-t003:** Factors correlated with higher endorsement of conspiracy beliefs about emerging virus infections among the study participants (*n* = 606).

Factors Associated with Higher Embrace of Conspiracy Beliefs about Emerging Virus Infections ^1^	Odds Ratio (95% Confidence Interval)	*p* Value
HMPX K-Score ^2^ ≤ 4 vs. HMPX K-Score > 4	1.496 (1.059–2.113)	0.022
Nurses vs. medical technicians	1.792 (0.999–3.213)	0.050
Physicians vs. medical technicians	0.495 (0.273–0.895)	0.020
Dentists vs. medical technicians	0.670 (0.333–1.346)	0.260
Pharmacists vs. medical technicians	0.852 (0.434–1.674)	0.642
Participants with agreement or neutral/no opinion belief in the role of male homosexuals in HMPX spread vs. those who disagreed	2.768 (1.764–4.345)	<0.001
Covariates		
Age < 34 years vs. ≥34 years	0.957 (0.665–1.377)	0.812
Males vs. females	0.719 (0.497–1.042)	0.081

^1^ The emerging virus infections conspiracy scale (EVICS) dichotomized as those with a score of EVICS ≥ 48 indicating a higher embrace of conspiracies vs. those with a score of <48 indicating a lower embrace of conspiracies with the former as the reference category. ^2^ HMPX K-Score: Human monkeypox knowledge score.

## Data Availability

The data presented in this study are available upon request from the corresponding author (M.S.).

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
