# Peer review of "Assessing Healthcare Workers’ Knowledge and Their Confidence in the Diagnosis and Management of Human Monkeypox: A Cross-Sectional Study in a Middle Eastern Country"

_healthcare, 2022, doi:10.3390/healthcare10091722_

Round 1
Reviewer 1 Report
Sallam et al report results of a cross-sectional study assessing the healthcare workers’ level of knowledge and their confidence in diagnosis and management of monkeypox in Jordan. The primary outcome variables were: (1) HMPX knowledge based in HMPX K-score, (2) confidence in HMPX diagnosis and management based in the confidence score, (3) attitude towards the role of MSM in the spread of HMPX and (4) conspiratorial attitude towards EID. 606 health workers completed an online self-administered questionnaire between May 2022 and July 2022 and were therefore included in the study. The study reports low level of HMPX knowledge among healthcare workers and it is associated with low confidence in management and diagnosis. There is a high embrace of conspiracy beliefs about emerging viral infections. Higher knowledge is observed among physicians compared to nurses and other healthcare workers.
The manuscript is well written, and I would recommend it for publication if some minor issues were addressed.
- You mentioned in the introduction that only three countries in the Middle East and 70 in North Africa have reported Monkeypox cases. Please, explain in the introduction section if any cases have been reported in Jordan and its implication in HMPX knowledge.
- A recent article published in The Lancet (PMID: 35952705) has strengthens the evidence of skin-to-skin contact during sex as the dominant mechanism of transmission of monkeypox, especially through sexual networks of MSM. Please add this reference in the introduction section.
- Is vaccination for the prevention of monkeypox among high-risk individuals or as a post-exposure prophylaxis of contacts available in Jordan? Please, comment on that in the discussion section.
Author Response
Reviewer #1 Comments and Suggestions for Authors
Sallam et al report results of a cross-sectional study assessing the healthcare workers’ level of knowledge and their confidence in diagnosis and management of monkeypox in Jordan. The primary outcome variables were: (1) HMPX knowledge based in HMPX K-score, (2) confidence in HMPX diagnosis and management based in the confidence score, (3) attitude towards the role of MSM in the spread of HMPX and (4) conspiratorial attitude towards EID. 606 health workers completed an online self-administered questionnaire between May 2022 and July 2022 and were therefore included in the study. The study reports low level of HMPX knowledge among healthcare workers and it is associated with low confidence in management and diagnosis. There is a high embrace of conspiracy beliefs about emerging viral infections. Higher knowledge is observed among physicians compared to nurses and other healthcare workers.
The manuscript is well written, and I would recommend it for publication if some minor issues were addressed.
Response: We are deeply thankful for the insightful summary and for the positive critical appraisal of the manuscript by the estimated reviewer.
- You mentioned in the introduction that only three countries in the Middle East and in North Africa have reported Monkeypox cases. Please, explain in the introduction section if any cases have been reported in Jordan and its implication in HMPX knowledge.
Response: Thanks for this important comment. So far, no HMPX cases has been confirmed in Jordan. In addition, as of 31 August 2022, the total number of HMPX cases in Arab countries of the Middle East and North Africa reached 30 that were distributed across five countries: United Arab Emirates (UAE): 16, Saudi Arabia: 8, Lebanon: 6, Morocco: 3, Qatar 3 and Sudan: 2 cases. This information is based on the 2022 Monkeypox Outbreak Global Map by the Centers for Disease Control and Prevention.
Based on the reviewer’s important suggestion, we added the following paragraph to the Introduction section (Page 2, lines 73-78): “By 31 August 2022, the total number of confirmed HMPX cases approached 50,000 in 99 countries/territories worldwide [8]. Thirty-eight HMPX cases have been confirmed in six Arab countries of the Middle East and North Africa (United Arab Emirates (UAE): 16, Saudi Arabia: 8, Lebanon: 6, Morocco: 3, Qatar: 3 and Sudan: 2 cases), with no recorded cases in Jordan by the end of August 2022 [8].”
- Centers for Disease Control and Prevention (CDC). 2022 Outbreak Cases and Data. Available online: https://www.cdc.gov/poxvirus/monkeypox/response/2022/index.html (accessed on 31 August 2022).
- A recent article published in The Lancet (PMID: 35952705) has strengthens the evidence of skin-to-skin contact during sex as the dominant mechanism of transmission of monkeypox, especially through sexual networks of MSM. Please add this reference in the introduction section.
Response: We are deeply grateful for highlighting this relevant and timely study by Eloy José Tarín-Vicente et al. We agree with the reviewer that this reference is a relevant addition to the Introduction in terms of transmission of MPXV, and variability in clinical presentation which is pertinent in clinical diagnosis. Accordingly, we added the suggested reference in the Introduction (Page 3, lines 102-104): “A recent study from Spain added further evidence supporting the hypothesis of skin-to-skin transmission of MPXV during sex as the dominant mode of transmission among MSM in the course of the ongoing outbreak [18].”
- Tarín-Vicente, E.J.; Alemany, A.; Agud-Dios, M.; Ubals, M.; Suñer, C.; Antón, A.; Arando, M.; Arroyo-Andrés, J.; Calderón-Lozano, L.; Casañ, C., et al. Clinical presentation and virological assessment of confirmed human monkeypox virus cases in Spain: a prospective observational cohort study. The Lancet 2022, 400,(10353): 661-669, doi:10.1016/S0140-6736(22)01436-2.
In addition, we added this influential reference to the following paragraphs in the Introduction: “The disease can be transmitted either by direct contact (with skin or mucous mem-brane lesions or via the respiratory tract/saliva), or indirectly through contaminated fomites [17,18].”; and “Although anyone in close contact with active cases can be at risk of virus acquisition, the current cases appear to be concentrated among men who have sex with men (MSM) [18,23-26].”
- Is vaccination for the prevention of monkeypox among high-risk individuals or as a post-exposure prophylaxis of contacts available in Jordan? Please, comment on that in the discussion section.
Response: Thanks for raising this important point. Regarding the availability of vaccination to prevent human monkeypox, the vaccine has not been made available in Jordan according to the Ministry of Health sources. Based on the reviewer’s suggestion, we added the following statement to the Discussion section (Page 10, lines 355-358): “Currently, vaccination to prevent HMPX is not available in Jordan [47]. However, vaccination might be needed soon at least for most-at-risk groups considering the spread of the virus to neighboring countries and at the global level. Thus, the assessment of HCWs’ attitude to HMPX vaccination is recommended soon.”
Reviewer 2 Report
The current manuscript urges a need for better education for HCWs in Jordan in facing with HMPX outbreak, especially for the possibility of smallpox vaccination. It also shows HCW might have specific attitude to conspiracy concept to HMPX or other infectious disease. The impact of the manuscript will be positive and helpful to Jordan in facing HMPX outbreak. It is however good to review the study for whether some parts might be oversimplified or overinterpreted. Possible parts are listed below:
1. In addition to age, it will be useful to provide details about the working nature of the participants such as seniority, specialty, career grading, or any quality that will affect their knowledge and management response to HMPX.
2. Some HMPX knowledge questions might not be very accurate:
a. The question design to “antibiotics are used to treat human monkeypox” might not be very accurate basing on the fact that most participants know “Monkeypox is caused by a virus”. It is still possible that antibiotics are useful for some situations such as secondary bacterial infection.
b. The question “There is an outbreak of human monkeypox in the world” is too subjective and how to define “world” can vary among different individuals.
3. It may be of bias to simply question “Monkeypox spread worldwide due to the role of male homosexuals” but without other choices such as heterosexual, female homosexuals or other transmission routes.
4. Finally, it will be useful to elaborate the limitation of the current study in 446-449 to facilitate potential follow-up studies. That includes discussion to study design, sample size and quality and study methodology.
Author Response
Reviewer #2 Comments and Suggestions for Authors
The current manuscript urges a need for better education for HCWs in Jordan in facing with HMPX outbreak, especially for the possibility of smallpox vaccination. It also shows HCW might have specific attitude to conspiracy concept to HMPX or other infectious disease. The impact of the manuscript will be positive and helpful to Jordan in facing HMPX outbreak.
Response: We are deeply thankful for the summary of the manuscript.
- It is however good to review the study for whether some parts might be oversimplified or overinterpreted. Possible parts are listed below: In addition to age, it will be useful to provide details about the working nature of the participants such as seniority, specialty, career grading, or any quality that will affect their knowledge and management response to HMPX.
Response: Thanks for raising this important issue. However, the survey did not include items that assessed seniority, specialty, career grading, and other relevant qualifications. We agree with the reviewer that this was an important point to consider, which unfortunately was not included in the survey instrument. Based on the reviewer’s comment, we acknowledge that such missing items were a limitation of the current study and should be addressed in future work. Therefore, we added the following statement to the limitations paragraph of the Discussion section (Page 12, lines 462-466): “Moreover, the current study was limited by the lack of items assessing seniority, specialty, career grading, and other relevant qualifications of HCWs’ categories similar to the approach taken by Matteo Riccò et al. [29]. Such items are needed in the future work aiming to discern differences within each occupational category in terms of HMPX knowledge and confidence levels in diagnosis and management of the disease.”
- Riccò, M.; Ferraro, P.; Camisa, V.; Satta, E.; Zaniboni, A.; Ranzieri, S.; Baldassarre, A.; Zaffina, S.; Marchesi, F. When a Neglected Tropical Disease Goes Global: Knowledge, Attitudes and Practices of Italian Physicians towards Monkeypox, Preliminary Results. Trop Med Infect Dis 2022, 7,(7), doi:10.3390/tropicalmed7070135.
- Some HMPX knowledge questions might not be very accurate:
- The question design to “antibiotics are used to treat human monkeypox” might not be very accurate basing on the fact that most participants know “Monkeypox is caused by a virus”. It is still possible that antibiotics are useful for some situations such as secondary bacterial infection.
- The question “There is an outbreak of human monkeypox in the world” is too subjective and how to define “world” can vary among different individuals.
Response: We thank the reviewer for raising these points. However, we respectfully disagree with this viewpoint based on the following: The knowledge items, as mentioned in the Methods section, were adopted from a previous study by Harapan et al. doi:10.1080/20477724.2020.1743037; the item that assessed the knowledge regarding the lack of utility of antibiotics in the treatment of monkeypox is of particular relevance in the Middle East, and particularly in Jordan. This comes in light of unsatisfactory knowledge about antimicrobial stewardship programs (doi:10.1093/jphsr/rmaa034), and major issues regarding dispensing of antibiotics without a prescription (doi:10.2147/PPA.S91649). Thus, we feel that the item was necessary to show the proportion of HCWs who believed in the value of antibiotics to treat monkeypox itself rather than its complications (including secondary bacterial infections). Regarding the item “There is an outbreak of human monkeypox in the world” we feel that it was clearly fair in Arabic language based on the content validity of the survey to assess the level of knowledge and awareness about the ongoing multi-country HMPX outbreak.
- It may be of bias to simply question “Monkeypox spread worldwide due to the role of male homosexuals” but without other choices such as heterosexual, female homosexuals or other transmission routes.
Response: We would like to thank the reviewer for raising this important point. We are aware that a single item to assess knowledge that HMPX spread (similar to any emerging and re-emerging virus infection) occur due to multiple factors rather than the single role of a group of individuals (even if they are considered at-risk group). However, we believe that the utility of this item can be useful as an initial point to conduct further studies probing the attitude of HCWs in the region towards patients with HMPX and particularly MSM with the disease. We believe that the following paragraph in the Discussion was a clear indication of the limited information drawn from the single item: “In this study, a majority of participants agreed that MSM had a role in the spread of the ongoing HMPX outbreak. This result was uniform across all sociodemographic and occupational categories. Although this finding does not necessarily imply a negative attitude towards MSM stricto sensu, it should be evaluated further especially in the Middle East region, where a few studies pointed to the generally negative attitude towards MSM in the context of HIV infection, besides the reporting of social stigma and bullying towards COVID-19 patients in the early phases of the pandemic.”
However, and based on the reviewer’s important comment, we added the following statement to the limitations paragraph of the Discussion section: “Furthermore, the evaluation of HCWs’ attitude towards the role of MSM in HMPX spread was based on a single item, with subsequent risk of measurement bias. Therefore, we encourage future studies focusing on the prevalence of HCWs’ stigmatizing attitude towards HMPX patients, and particularly towards most-at-risk groups using a more rigorous approach.”
- Finally, it will be useful to elaborate the limitation of the current study in 446-449 to facilitate potential follow-up studies. That includes discussion to study design, sample size and quality and study methodology.
Response: We are thankful for this comment. Accordingly, we updated the limitations part of the Discussion section as follows (Page 12, lines 458-472): “The study limitations included selection bias based on the sampling approach, with a limited sample size besides the inherent social desirability bias in relation to EVICS items. In addition, the future studies should assess the attitude towards vaccination and the possible stigmatizing attitude towards MSM considering their high risk of virus acquisition [72]. Moreover, the current study was limited by the lack of items assessing seniority, specialty, career grading, and other relevant qualifications of HCWs’ categories similar to the approach taken by Matteo Riccò et al. [29]. Such items are needed in the future work aiming to discern differences within each occupational category in terms of HMPX knowledge and confidence levels in diagnosis and management of the disease. Furthermore, the evaluation of HCWs’ attitude towards the role of MSM in HMPX spread was based on a single item, with subsequent risk of measurement bias. Therefore, we encourage future studies focusing on the prevalence of HCWs’ stigmatizing attitude towards HMPX patients, and particularly towards most-at-risk groups using a more rigorous approach. Finally, addressing the potential hesitancy towards HMPX vaccination and its associated determinants should be considered in future studies [29].”
- Riccò, M.; Ferraro, P.; Camisa, V.; Satta, E.; Zaniboni, A.; Ranzieri, S.; Baldassarre, A.; Zaffina, S.; Marchesi, F. When a Neglected Tropical Disease Goes Global: Knowledge, Attitudes and Practices of Italian Physicians towards Monkeypox, Preliminary Results. Trop Med Infect Dis 2022, 7,(7), doi:10.3390/tropicalmed7070135.
- Al Awaidy, S.T.; Khamis, F.; Sallam, M.; Ghazy, R.M.; Zaraket, H. Monkeypox Outbreak: More queries posed as cases globally soar. Sultan Qaboos University Medical Journal 2022, 1,(1): n/a, doi:10.18295/squmj.8.2022.046.
Round 2
Reviewer 2 Report
Study limitation and plans are adequately provided. No further review is required.